# Postprandial Hypotension as a Risk Factor for the Development of New Cardiovascular Disease: A Prospective Cohort Study with 36 Month Follow-Up in Community-Dwelling Elderly People

**DOI:** 10.3390/jcm9020345

**Published:** 2020-01-27

**Authors:** Aelee Jang

**Affiliations:** Department of Nursing, University of Ulsan, 93, Daehak-ro, Nam-gu, Ulsan 44610, Korea; aeleejang@ulsan.ac.kr; Tel.: +82-52-259-1252

**Keywords:** cardiovascular disease, postprandial, hypotension, blood pressure, elderly

## Abstract

Postprandial hypotension (PPH) is common among the elderly. However, it is unknown whether the presence of PPH can predict the development of new cardiovascular disease (CVD) in the elderly during the long-term period. This study aimed to prospectively evaluate the presence of PPH and the development of new CVD within a 36 month period in 94 community-dwelling elderly people without a history of CVD. PPH was diagnosed in 47 (50.0%) participants at baseline and in 7 (7.4%) during the follow-up period. Thirty participants (31.9%) developed new CVD within 36 months. We performed a time-dependent Cox regression analysis with PPH, hypertension, diabetes, and body mass index (BMI) as time-varying covariates. In the univariate analyses, the presence of PPH, higher BMI, hypertension, diabetes mellitus, and higher systolic and diastolic blood pressure were associated with the development of new CVD. The multivariate analysis indicated that the relationship between PPH and the development of new CVD remained (adjusted hazard ratio 11.18, 95% confidence interval 2.43–51.38, *p* = 0.002) even after controlling for other variables as covariates. In conclusion, the presence of PPH can predict the development of new CVD. Elderly people with PPH may require close surveillance to prevent CVD.

## 1. Introduction

Postprandial hypotension (PPH) is a common but often unrecognized disorder in the elderly [1]. PPH is defined as a ≥20 mmHg decrease in systolic blood pressure (SBP) within 2 h after a meal [2]. The prevalence of PPH is 20–91% in hospitalized geriatric patients [3,4,5,6]. The pathogenesis of PPH is unclear, but an inadequate cardiovascular response to postprandial splanchnic blood pooling is regarded as a primary mechanism [2]. In the elderly, sympathetic activation to decrease the effective circulating volume is often suppressed; hence, a continuous pooling of blood in the splanchnic bed may result in a significant decrease in blood pressure (BP) after a meal [7,8]. Changes in cardiovascular function and the neurohormonal response with aging can also contribute to diverse comorbidities associated with the dysregulation of BP homeostasis [9].

Cardiovascular diseases (CVDs), such as stroke, transient ischemic attack, angina, and myocardial infarction, are the main cause of mortality and work disability in the elderly [10,11]. Early recognition of subclinical risk factors is essential for the proper surveillance and prevention of new CVD. PPH is also recognized as an important clinical issue, particularly as a significant risk factor for subsequent CVD, but this has not yet been accurately confirmed [2,12]. Some studies have suggested an association between PPH and CVD and mortality [9,13]. However, it is difficult to determine the casual relationship between PPH and CVD development owing to differences in patient selection, study design, and diagnosis criteria among previous studies [9,14,15,16]. Furthermore, the diagnosis of PPH was heterogeneous in previous studies, such as with regard to the time interval between meal consumption and BP measurement [5]. Therefore, to investigate whether the presence of PPH can predict the development of new CVD in the elderly, this study aimed to prospectively evaluate the existence of PPH and the development of new CVD among community-dwelling elderly people in a period of 36 months.

## 2. Materials and Methods

### 2.1. Study Design and Participants

This prospective cohort study enrolled 94 participants from three senior community centers in South Korea between 2011 and 2015. The inclusion criteria were (1) age ≥65 years, (2) ability to eat independently and maintain a sitting position for 2 h after a meal, and (3) ability to perform activities of daily living independently. The exclusion criteria included (1) impaired cognitive function, (2) psychiatric illness, (3) recent hospitalization owing to acute illness within 1 month prior to the study, (4) medical history of CVD (congestive heart failure, myocardial infarction, angina, or cerebrovascular accidents, including transient ischemic attack), or (5) use of medications affecting gastrointestinal motility. After evaluating the presence of PPH at baseline, the new onset of PPH was evaluated during the follow-up period of 36 months. Body mass index (BMI) and the presence of hypertension and diabetes were also assessed regularly. This study was conducted following the guidelines of the Declaration of Helsinki of 1975, revised in 2013, and was approved by the institutional review board of Pusan National University Hospital (E-2014007), located in B city, South Korea. All participants provided written informed consent.

### 2.2. Evaluation of PPH and Acquisition of Demographic Data

BP was measured using an automated sphygmomanometer (A&D UA-851; A&D Company, Japan) following the European Society of Hypertension guidelines [17]. BP was measured by a well-trained nurse with the participant in a sitting position. Baseline BP was measured before a meal. Participants were asked to refrain from eating food, drinking coffee or alcohol, and smoking 4 h before measurement. All participants took any prescription medications after the evaluation of PPH, including antihypertensive drugs and hypoglycemic agents. BP was measured twice with a 5 min interval before a meal, and the average of the two measurements was used as the baseline BP. To measure postprandial BP, participants were provided the same standardized meal comprising 210 g rice, 100 g soup, and 70 g side dishes, which was approximately 500 kcal. After the participant consumed the meal, postprandial BP was measured every 15 min for 2 h. PPH was defined as a decrease in SBP of ≥20 mmHg from the baseline within 2 h after a meal [2]. We analyzed the reliability of measurements and found that the intraclass correlation coefficients (ICCs) for intra-rater reliability for SBP and diastolic blood pressure (DBP) were 0.995 (95% confidence interval (CI) 0.975–0.998, *p* < 0.001) and 0.989 (95% CI 0.942–0.996, *p* < 0.001), respectively.

Demographic and clinical information was also obtained by interviewing the participants using a questionnaire. Information on age, living conditions, alcohol intake, and smoking habits was collected, and information on medical history was obtained from hospital records after consent was obtained from the participants and their families. Bodyweight and height were measured using an automated measuring machine (G-Tech International Co., Ltd., Uijeongbu, Korea) to determine the BMI.

### 2.3. Surveillance of CVD Development

The primary endpoint was the occurrence of new CVD within 36 months. CVD in this study included coronary heart disease (such as congestive heart failure, angina, and myocardial infarction) and cerebrovascular disease (such as stroke and transient ischemic attack) according to the World Health Organization definition of CVD [18]. Participants were regularly followed every 3 months, and CVD-related information was obtained from them or their families during their visits to the centers or via telephone calls. If the participant was diagnosed with a new disease, additional information was also obtained from the medical record or prescription from the hospital at which the participant was diagnosed with new conditions, including the new onset of hypertension, diabetes, and CVDs.

### 2.4. Statistical Analysis

The reliability of BP measurements was calculated by determining the ICC (two-way random effects model). Variables are expressed as frequencies and percentages for categorical data, and means ± standard deviation for numerical data. Group differences were assessed using the chi-squared test or Fisher’s exact test for categorical data and the independent t test or Mann–Whitney U test for numerical data as appropriate. To determine whether the distribution was normal, we used the Shapiro–Wilk test. Time to CVD was estimated using Kaplan–Meier curves. Survival curves were compared between groups using the log-rank test. Cox regression models with time-varying covariates were used to identify prognostic factors that were independently related to CVD. The main predictors were baseline PPH and PPH as a time-varying covariate (incorporating new onset PPH during the follow-up and duration of PPH as a time-varying covariate). In addition, hypertension, diabetes mellitus, and BMI were monitored during the follow-up period. These predicting variables were considered as time-varying covariates in time-dependent Cox regression models. All statistical analyses were performed using SPSS version 24.0 (IBM Corp., Armonk, NY, USA) and R version 3.5.1, and *p*-values <0.05 were considered statistically significant.

## 3. Results

### 3.1. Baseline Characteristics

The participants comprised 79 (84.0%) females and 15 (16.0%) males. The mean age was 73.1 ± 4.8 years, and the mean BMI was 23.7 ± 2.5 kg/m^2^. Among the participants, 67 (71.3%) had lower than middle school education, and 62 (66.0%) were currently living with their family or spouse. The most common comorbidity was hypertension (50.0%), followed by diabetes mellitus (19.1%). At baseline, 47 (50%) participants were diagnosed with PPH, but there were no significant differences in baseline characteristics, with the exception of BP, between participants with and without PPH (Table 1).

### 3.2. Incidence of CVD

During the follow-up, 30 (31.9%) patients developed new CVD, of whom 4 passed away owing to acute myocardial infarction (*n* = 2) and ischemic stroke (*n* = 2). The CVD incidence was significantly higher in the group with PPH than the group without PPH at baseline (55.3% vs. 8.5%, *p* < 0.001, Figure 1).

Table 2 indicates the differences between groups with respect to CVD development. The presence of PPH at baseline (86.7% in the CVD group vs. 32.8% in the non-CVD group, *p* < 0.001), higher BMI (24.5 ± 2.1 in the CVD group vs. 23.3 ± 2.7 in the non-CVD group), hypertension (66.7% in the CVD group vs. 42.2% in the non-CVD group), higher baseline SBP (139.3 ± 20.9 mmHg in the CVD group vs. 123.6 ± 17.9 mmHg in the non-CVD group), and higher baseline DBP (79.0 ± 8.6 mmHg in the CVD group vs. 73.4 ± 10.1 mmHg in the non-CVD group) were significantly related to the development of new CVD. The percentage of participants with an SBP change of ≥20 mmHg (86.7% vs. 32.8%, respectively, *p* < 0.001) and a DBP change of ≥10 mmHg (73.3% vs. 48.4%, respectively, *p* = 0.023) were significantly higher among participants who developed new CVD than among those who did not.

### 3.3. Predictive Factors for the Development of New CVD

Ninety-four patients were available for the Cox regression models to assess the relationship between PPH and CVD using PPH as the time-varying covariate. There were 30 cases of CVD. We considered time-varying covariates, including PPH, hypertension, diabetes, and BMI, in the time-dependent Cox regression models. Among the 47 patients without PPH at baseline, 7 were diagnosed with PPH during the follow-up period. The main predictive factors were baseline PPH, with PPH as a time-varying covariate incorporating new onset PPH during follow-up, and the duration of PPH as a time-varying covariate. New onset hypertension and diabetes mellitus during the follow-up period were observed in 12 and 11 patients, respectively. In addition, BMI was measured at baseline, 1 year, and 2 years. All predictive variables were considered as time-varying covariates in time-dependent Cox regression models. In the univariate analyses, the presence of PPH, higher BMI, hypertension, diabetes mellitus, and higher SBP and DBP were found to be associated with the development of new CVD, while age, sex, education level, living status, alcohol consumption, and smoking were not. Patients with PPH were more likely to develop new CVD (crude hazard ratio (HR) 15.97, 95% CI 3.80–67.08, *p* < 0.001). The multivariate analysis that included significant factors in the univariate analyses as covariates revealed that the relationship between PPH and the development of new CVD remained (adjusted HR 11.18, 95% CI 2.43–51.38, *p* = 0.002) even after controlling for other variables as covariates in the multivariate analysis. Hypertension was also found to be a significant factor affecting CVD (adjusted HR 3.26, 95% CI 1.22–8.76, *p* = 0.019) (Table 3).

## 4. Discussion

This study investigated the time-varying effect of PPH associated with new CVD in community-dwelling elderly people. The study indicated that approximately half (55.3%) of the participants with PPH developed new CVD within the 36 month follow-up period, whereas only 8.5% of participants without PPH developed CVD. After adjustment for other covariates, the time-varying effect of PPH on the development of new CVD was found to be significant (HR 11.18, 95% CI 2.43–51.38). These results suggest that elderly people with PPH were more likely to develop new CVD and, thus, required close surveillance.

The reported prevalence of PPH varies widely in previous studies depending on the patient population and diagnostic methods [5]. Studies of institutionalized elderly patients reported a PPH prevalence of 24–38% [3,4]. Meanwhile, the incidence of PPH among hospitalized geriatric patients was considerably high (up to 91%) [5,6]. However, studies investigating the incidence of PPH among healthy elderly people are relatively rare [19]. Unlike previous studies, the present study investigated PPH in relatively healthy elderly people who were not institutionalized or hospitalized. Although the study population comprised elderly patients, the PPH incidence was 50%, which seems to be slightly higher than that reported in previous studies. The development of PPH is highly dependent on meal composition. Carbohydrate-rich meals predispose patients to a more immediate decrease in BP than do meals containing primarily protein or fat [20]. Therefore, it is expected that PPH might be more prevalent in people from Asian regions owing to their carbohydrate-dominant diets. However, few studies have investigated PPH in Asian patients [21]. To date, this is the first prospective cohort study to investigate PPH and its association with CVD in relatively healthy Asian elderly people. In this study, a stringent diagnostic approach was used for PPH, such as provision of a standardized meal and more frequent BP measurements. Furthermore, in this study, variance in BP changes related to food composition was reduced by administering the same standardized meal to all participants. Therefore, this study may provide relevant insights into the epidemiology of PPH in Asian regions. Large cohort studies are recommended to further determine the prevalence of PPH in different patient populations using a standardized diagnostic method.

In this study, 30 (31.9%) participants developed new CVD during the 36 month follow-up. In one study on PPH as a predictor of new CVD [14], 40.8% of older nursing home residents developed CVD during a 29 month follow-up. The patients with new CVD exhibited a significantly greater decline in postprandial SBP than did those without (*p* < 0.001). The other previous study found a 59.2% occurrence of new CVD during a 4.7 year follow-up period in older low-level-care residents in long-term health facilities [9]. The occurrence of new CVD cases was lower in this study (31.9%) than in previous studies (40.1–59.2%) because this study was conducted in a relatively healthy cohort from a community setting. Furthermore, the previous studies did not fully exclude populations with a history of CVD. In contrast, this prospective study excluded people with a history of CVD to clearly demonstrate the relationship between PPH and CVD.

In this study, the presence of PPH, higher BMI, hypertension, diabetes mellitus, and higher SBP and DBP were predictive factors for the development of CVD in the univariate analysis using the time-dependent Cox proportional hazard model. After adjustment for these factors, the time-varying effect of PPH and hypertension on the occurrence of new CVD still remained in the elderly people. According to Aronow et al. [14], the mean maximal decrease in postprandial SBP was a significant risk factor for developing CVD. Similarly, in previous cross-sectional studies, PPH was considered an independent predictor of asymptomatic cerebrovascular damage in healthy participants and hospitalized patients with essential hypertension [15,16]. Despite the small sample size, this study is meaningful as a prospective cohort study for the risk factors of new CVD in the presence or absence of PPH among community-dwelling elderly people without a history of CVD.

The total mortality (7/94, 7.4%) and CVD-related mortality (4/94, 4.3%) rates in this study were relatively lower than those reported in previous studies (16.9–54.2%) [9,13,14]. Again, this is likely because this study included relatively healthy older adults (i.e., from a community setting) in contrast to previous studies, which generally included hospitalized geriatric patients or patients in nursing homes. The mean age of the participants in this study (73.1 years) was also relatively lower than that in previous studies (77.8–83.2 years) [9,13,14]. In a previous study of hypertensive elderly patients who were followed up at the cardiology clinic for 4 years, the total mortality rate was 16.9%, and a greater decrease in postprandial SBP was associated with a greater increase in CVD mortality. In that prospective study, CVD mortality constituted 50% of the total mortality rate, while PPH accounted for 52.9% of deaths owing to CVD [13], which was similar to the findings of the present study. In a study by Fisher et al. [9], PPH was the only risk factor for all-cause mortality, and 50% of deaths were attributed to CVD. Participants with a postprandial SBP difference of ≥20 mmHg had the lowest survival during the 4.7 year follow-up period. They also reported that PPH accounted for 47% of deaths in long-term healthcare facilities, which was lower than the 57.1% (4 of 7 deaths) found in the present study. Populations in long-term healthcare facilities typically require nursing care for monitoring and preventing serious diseases, including CVD. These perspectives may explain the difference in attributing the total mortality rate to CVD mortality between the present study and the previous study mentioned above [9]. Therefore, prevention and close monitoring are needed to reduce deaths due to new CVD by classifying patients diagnosed with PPH as high risk for CVD.

This study had some limitations. First, the sample size was relatively small. However, compared to previous studies, this study was conducted over a long-term follow-up with the time-varying effect of PPH. The post hoc power for investigating the primary objective of this study was approximately 99%. Second, despite efforts to recruit a similar ratio of males and females, the participants were predominantly female because of the higher proportion of elderly women in the Korean elderly population. Future studies should confirm CVD risk factors in sex-balanced populations with postprandial SBP reduction during long-term follow-up. Third, the level of physical activity was not considered in this study, which could affect the development of CVDs. Nevertheless, the participants had similar lifestyle patterns because they resided in the same town and enjoyed the same activity programs provided by the community welfare center. Therefore, their physical activity levels may be similar. Fourth, the pathophysiological relationship between PPH and CVD remains vague despite PPH being identified as an independent time-varying variable for the development of new CVD. Autonomic dysfunction may play a major role in the development of PPH-related new CVD because PPH has been reported to be pathologically associated with autonomic dysfunction. This needs further evaluation using animal models. Fifth, the positive association between PPH and CVD may only reflect the high baseline SBP on the risk of CVD because the baseline SBP level in the PPH group was much higher than that in the non-PPH group. Although hypertension was adjusted in the multivariate model, based on the small sample size, the effect of high baseline SBP may persist. As it is impossible to stratify the analysis by hypertension status at baseline, owing to the limited sample size, further research in the form of a well-designed, large-sample study in a normotensive elderly population is necessary to definitively establish the effect of PPH on the risk of CVD.

This study investigated the relationship between PPH and the development of new CVD in elderly people during a long-term follow-up period. The results also suggest that the underlying reason of CVD development could be the greater decline in postprandial SBP (such as that occurring in PPH). This study demonstrated that increases in SBP variations after a meal may be a manifestation of CVD development and may reflect the presence of subclinical cardiovascular damage.

## 5. Conclusions

This study revealed that PPH was an independent predictor of new CVD among community-dwelling elderly people during a 36 month follow-up. Thus, CVD development may be prevented or monitored based on the presence of PPH. However, longitudinal studies in larger samples are needed to clarify the prognostic significance of PPH in the development of new CVD.

## Figures and Tables

**Figure 1 jcm-09-00345-f001:**
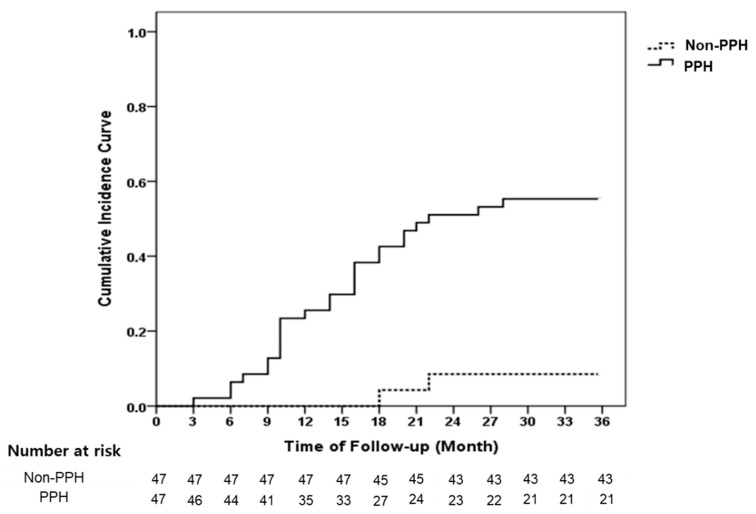
Kaplan–Meier curve of the incidence of new CVD between participants with and without PPH at baseline. The 3 year incidence of CVD was significantly higher in the group with PPH than in the group without PPH at baseline (55.3% vs. 8.5%, log-rank test, *p* < 0.001). PPH, postprandial hypotension; CVD, cardiovascular disease.

**Table 1 jcm-09-00345-t001:** Baseline characteristics of participants with and without PPH.

Characteristics	Total (*n* = 94)	PPH (*n* = 47)	Non-PPH (*n* = 47)	*p*-Value
Age (years)	73.1 ± 4.8	73.6 ± 4.4	72.7 ± 5.1	0.355 ^1^
Sex				0.778 ^3^
Male	15 (16.0)	8 (17.0)	7 (14.9)	
Female	79 (84.0)	39 (83.0)	40 (85.1)	
Body Mass Index (kg/m^2^)	23.7 ± 2.5	23.8 ± 2.7	23.5 ± 2.4	0.642 ^2^
Education Level				0.509 ^3^
Elementary School	67 (71.3)	36 (76.6)	31 (66.0)	
Middle School	13 (13.8)	5 (10.6)	8 (17.0)	
High School	14 (14.9)	6 (12.8)	8 (17.0)	
Living Status				1.000 ^3^
Living Alone	32 (34.0)	16 (34.0)	16 (34.0)	
With Family or Spouse	62 (66.0)	31 (66.0)	31 (66.0)	
Alcohol Drinking				0.370 ^3^
Yes	13 (13.8)	5 (10.6)	8 (17.0)	
No	81 (86.2)	42 (89.4)	39 (83.0)	
Smoking				1.000 ^4^
Yes	4 (4.3)	2 (4.3)	2 (4.3)	
No	90 (95.7)	45 (95.7)	45 (95.7)	
Hypertension	47 (50.0)	26 (55.3)	21 (44.7)	0.302 ^3^
Diabetes Mellitus	18 (19.1)	11 (23.4)	7 (14.9)	0.294 ^3^
Baseline SBP (mmHg)	128.6 ± 20.2	140.1 ± 17.2	117.1 ± 16.1	<0.001 ^2^
<120	34 (36.2)	6 (12.8)	28 (59.6)	<0.001 ^3^
≥120 to <140	31 (33.0)	15 (31.9)	16 (34.0)	
≥140	29 (30.9)	26 (55.3)	3 (6.4)	
Baseline DBP (mmHg)	75.2 ± 9.9	78.7 ± 7.5	71.7 ± 10.9	<0.001 ^2^
<80	64 (68.1)	23 (48.9)	41 (87.2)	<0.001 ^3^
≥80	30 (31.9)	24 (51.1)	6 (12.8)	
Postprandial SBP Change (mmHg)				<0.001 ^3^
<10	26 (27.7)	0 (0.0)	26 (55.3)	
≥10 to <20	21 (22.3)	0 (0.0)	21 (44.7)	
≥20	47 (50.0)	47 (100.0)	0 (0.0)	
Postprandial DBP Change (mmHg)				<0.001 ^3^
<10	41 (43.6)	9 (19.1)	32 (68.1)	
≥10	53 (56.4)	38 (80.9)	15 (31.9)	

Values are either frequency with percentage in parentheses or mean ± standard deviation. PPH, postprandial hypotension; SBP, systolic blood pressure; DBP, diastolic blood pressure. ^1^ P values were derived from independent t tests. ^2^ P values were derived from Mann–Whitney’s U test. ^3^ P values were derived using chi-square tests. ^4^ P values were derived using Fisher’s exact test. Shapiro–Wilk’s test was employed for test of normality assumption.

**Table 2 jcm-09-00345-t002:** Differences of characteristics between participants who developed and did not develop new CVD.

	Total (*n* = 94)	CVD (*n* = 30)	Non-CVD (*n* = 64)	*p*-Value
PPH				<0.001 ^3^
Yes	47 (50.0)	26 (86.7)	21 (32.8)	
No	47 (50.0)	4 (13.3)	43 (67.2)	
Age (years)	73.1 ± 4.8	73.0 ± 4.1	73.2 ± 5.1	0.849 ^1^
Sex				1.000 ^4^
Male	15 (16.0)	5 (16.7)	10 (15.6)	
Female	79 (84.0)	25 (83.3)	54 (84.4)	
Body mass index (kg/m^2^)	23.7 ± 2.5	24.5 ± 2.1	23.3 ± 2.7	0.013 ^2^
Education level				0.534 ^4^
Elementary school	67 (71.3)	24 (80.0)	43 (67.2)	
Middle school	13 (13.8)	3 (10.0)	10 (15.6)	
High school	14 (14.9)	3 (10.0)	11 (17.2)	
Living status				0.571 ^3^
Living alone	32 (34.0)	9 (30.0)	23 (35.9)	
With family or spouse	62 (66.0)	21 (70.0)	41 (64.1)	
Alcohol drinking				0.336 ^4^
Yes	13 (13.8)	6 (20.0)	7 (10.9)	
No	81 (86.2)	24 (80.0)	57 (89.1)	
Smoking				0.590 ^4^
Yes	4 (4.3)	2 (6.7)	2 (3.1)	
No	90 (95.7)	28 (93.3)	62 (96.9)	
Hypertension	47 (50.0)	20 (66.7)	27 (42.2)	0.027 ^3^
Diabetes mellitus	18 (19.1)	9(30.0)	9 (14.1)	0.067 ^3^
Baseline SBP (mmHg)	128.6 ± 20.2	139.3 ± 20.9	123.6 ± 17.9	<0.001 ^2^
<120	34 (36.2)	5 (16.7)	29 (45.3)	<0.001 ^3^
≥120 to <140	31 (33.0)	7 (23.3)	24 (37.5)	
≥140	29 (30.9)	18 (60.0)	11 (17.2)	
Baseline DBP (mmHg)	75.2 ± 9.9	79.0 ± 8.6	73.4 ± 10.1	0.005 ^2^
<80	64 (68.1)	15 (50.0)	49 (76.6)	0.010 ^3^
≥80	30 (31.9)	15 (50.0)	15 (23.4)	
Postprandial SBP change (mmHg)				<0.001 ^3^
<10	26 (27.7)	3 (10.0)	23 (35.9)	
≥10 to <20	21 (22.3)	1 (3.3)	20 (31.3)	
≥20	47 (50.0)	26 (86.7)	21 (32.8)	
Postprandial DBP change (mmHg)				0.023 ^3^
<10	41 (43.6)	8 (26.7)	33 (51.6)	
≥10	53 (56.4)	22 (73.3)	31 (48.4)	

Values are either frequency with percentage in parentheses or mean ± standard deviation. PPH, postprandial hypotension; CVD, cardiovascular disease; SBP, systolic blood pressure; DBP, diastolic blood pressure. ^1^ P values were derived from independent t tests. ^2^ P values were derived from Mann–Whitney’s U test. ^3^ P values were derived using chi-square tests. ^4^ P values were derived using Fisher’s exact test. Shapiro–Wilk’s test was employed for test of normality assumption.

**Table 3 jcm-09-00345-t003:** Summary of time-dependent Cox regression analyses using time-varying covariates (a total of 94 subjects, 30 CVDs).

Characteristic	Univariate			Multivariate		
	HR	95% CI	*p*-Value	HR	95% CI	*p*-Value
PPH	15.97	3.80–67.08	<0.001	11.18	2.43–51.38	0.002
Age (years)	1.00	0.93–1.07	0.940			
Sex (female)	0.90	0.34–2.35	0.830			
BMI (kg/m^2^)	1.14	1.01–1.29	0.030	1.10	0.96–1.26	0.173
Education level (vs. Elementary school)						
Middle school	0.61	0.19–2.04	0.430			
High school	0.57	0.17–1.88	0.350			
Living status (Living alone)	0.80	0.37–1.75	0.580			
Alcohol drinking	1.61	0.66–3.95	0.300			
Smoking	1.77	0.42–7.46	0.430			
Hypertension	4.61	1.77–12.05	0.002	3.26	1.22–8.73	0.019
Diabetes mellitus	3.20	1.56–6.56	0.002	1.80	0.84–3.89	0.132
Baseline SBP (mmHg)	1.03	1.02–1.05	<0.001	1.02	0.99–1.05	0.148
Baseline DBP (mmHg)	1.03	1.01–1.06	0.013	0.97	0.91–1.02	0.245

HR, hazard ratio; CVD, cardiovascular disease; BMI, body mass index; SBP, systolic blood pressure; DBP, diastolic blood pressure; PPH, postprandial hypotension.

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
