# Peer review of "Postprandial Hypotension as a Risk Factor for the Development of New Cardiovascular Disease: A Prospective Cohort Study with 36 Month Follow-Up in Community-Dwelling Elderly People"

_jcm, 2020, doi:10.3390/jcm9020345_

Round 1

Reviewer 1 Report

In my comment last time, I suggested the authors provide the baseline blood pressure level (SBP and DBP) in PPH and Non-PPH groups in mean (SD) format. The authors reported these values in this revised version. This also indicates a key issue in this paper.

From Table 1, the mean (SD) of SBP in the PPH group is 140.1±17.2 mmHg, while this value in the Non-PPH group is 117.1±16.1 mmHg. This means the majority of participants in PPH group are hypertension patients, while most people in Non-PPH group are normal people without hypertension.

Hypertension is the most well-known risk factor of heart disease, and it’s possible that people with hypertension at baseline are more likely to experience PPH. The association observed between PPH and incident CVD may only reflect the role of hypertension (or high baseline blood pressure level) on risk of CVD. Although the authors adjusted hypertension in the multivariate model, based on the small sample size, I suppose some effect of baseline blood pressure level is still there. This point needs to be acknowledged in the limitation part.

An appropriate design is to separate the people into two groups: hypertension group, and non-hypertension group.

In each group to examine whether people experience PPH increase the risk of CVD.

i.e., in hypertension group, to see whether people have PPH increased their risk of CVD, compared with people without PPH, adjusting baseline blood pressure level.

In non-hypertension group, to see whether people have PPH increased their risk of CVD, compared with people without PPH, adjusting baseline blood pressure level.

If the authors have no enough sample size now, this point can be included in the discussion point as a future research needed.

Minor comments:

What is “bPPH” in Table 1? Is it typo? Put the contents in lines 160-166 in Method part.

Author Response

Reviewer 1

Comments and Suggestions for Authors

In my comment last time, I suggested the authors provide the baseline blood pressure level (SBP and DBP) in PPH and Non-PPH groups in mean (SD) format. The authors reported these values in this revised version. This also indicates a key issue in this paper. From Table 1, the mean (SD) of SBP in the PPH group is 140.1±17.2 mmHg, while this value in the Non-PPH group is 117.1±16.1 mmHg. This means the majority of participants in PPH group are hypertension patients, while most people in Non-PPH group are normal people without hypertension. Hypertension is the most well-known risk factor of heart disease, and it’s possible that people with hypertension at baseline are more likely to experience PPH. The association observed between PPH and incident CVD may only reflect the role of hypertension (or high baseline blood pressure level) on risk of CVD. Although the authors adjusted hypertension in the multivariate model, based on the small sample size, I suppose some effect of baseline blood pressure level is still there. This point needs to be acknowledged in the limitation part.

An appropriate design is to separate the people into two groups: hypertension group, and non-hypertension group. In each group to examine whether people experience PPH increase the risk of CVD. i.e., in hypertension group, to see whether people have PPH increased their risk of CVD, compared with people without PPH, adjusting baseline blood pressure level, and in non-hypertension group, to see whether people have PPH increased their risk of CVD, compared with people without PPH, adjusting baseline blood pressure level. If the authors have no enough sample size now, this point can be included in the discussion point as a future research needed.

<Answer>

I agree with you. High baseline blood pressure in PPH group was my concern as well. I think that hypertension and PPH may partially share the pathophysiologic mechanism in elderly. Therefore, it is difficult to define whether the development of CVD is due to hypertension or PPH. For this reason, we performed sub-group analyses dividing into hypertension group and non-hypertension group to see whether PPH increase the risk of new CVD in both groups. In hypertension group, PPH is still found to be a significant factor affecting new CVD even after adjusting blood pressure at baseline. However, adjusted OR of PPH was not estimable in non-hypertension group due to the small sample size (See below).

Characteristic

Hypertension group

Non-hypertension group

HR

95% CI

p-Value

HR

95% CI

p-Value

PPH

11.90

2.13-66.51

0.005

592,988,700

0.00 – Inf

1.000

Baseline SBP (mmHg)

1.02

0.98-1.05

0.327

1.01

0.96-1.05

0.760

Baseline DBP (mmHg)

0.94

0.87-1.02

0.145

1.02

0.96-1.09

0.550

The reason for infinite odds ratio was that there was no patient developing new CVD in non-PPH patients in non-hypertension group (See below).

In non-hypertension group only

CVD (n=10)

Non-CVD (n=37)

PPH

Yes

10 (100.0)

11 (29.7)

no

0 (0.0)

26 (70.3)

Therefore, I added this limitation in Discussion (please, see lines 259-265) as follow:

Fifth, the positive association between PPH and CVD may be potentially attributed to hypertension, which is the best-known risk factor of CVD. At baseline, study participants with PPH had higher blood pressure than those without PPH. It is still possible that high baseline blood pressure may have affected the development of CVD although hypertension was adjusted in the multivariate analysis. Thus, further research in the form of a well-designed, large-sample study in a normotensive elderly population is necessary to definitively establish the effect of PPH on the risk of CVD.

Minor comments:

What is “bPPH” in Table 1? Is it typo? Put the contents in lines 160-166 in Method part.

This means baseline postprandial hypotension. I suppose to emphasize the baseline. I am sorry to make you confused. I changed bPPH to PPH or PPH at baseline.

Reviewer 2 Report

As described previously, the sample size was still very limited. Therefore, the statistical analyses in this study were insufficient. This is very important points. The increase in number of the subjects should be essential in this study.

Author Response

Reviewer 2

Comments and Suggestions for Authors

As described previously, the sample size was still very limited. Therefore, the statistical analyses in this study were insufficient. This is very important points. The increase in number of the subjects should be essential in this study.

<Answer>

Yes, I agree with you. We will get the increase in number of participants in future trial. As this is a prospective observational study, unfortunately it is not possible to increase the sample size at the moment. Even though we enroll more study participants now, we have to follow up 3 years from now and then study duration and clinical research nurse will be different which evokes some problematic biases as well. As another reviewer’s recommendation, we will evaluate the association between PPH and CVD in elderly without hypertension.

This study is limited because of small sample size. We described this limitation in Discussion. Nevertheless, this study is meaningful in several aspects including long-term study period, and time-varying effect of PPH. Furthermore, the post-hoc power to investigate the primary objective of this study was approximately 99%. Statistical analysis was carried out using R 3.5.1 software (See below).

Usage

powerCT.default0(k, m, RR, alpha = 0.05)

Arguments

Arguments

Definition

Value

k

ratio of participants in group E (experimental group) compared to group C (control group)

1

m

expected total number of events over both groups

30

RR

postulated hazard ratio

11.18

alpha

type I error rate

0.05

Results

> library (powerSurvEpi)

> powerCT.default0 (1, 30, 11.18, 0.05)

 [1] 0.9955761

Round 2

Reviewer 1 Report

I have no further comments. Based on the limited sample size, the authors did not have enough power to stratify the association by hypertension status.

If this paper was considered for publication, as I mentioned in my last comments, two points must be acknowledged clearly in the discussion part: (1) Because the baseline SBP level in the PPH group was much higher than that in the Non-PPH group, the association observed between PPH and incident CVD may only reflect the high baseline SBP on risk of CVD. Although the authors adjusted hypertension in the multivariate model, based on the small sample size, I suppose some effect of baseline blood pressure level is still there. (2) Due to the limited sample size, the authors have no ability to stratify the analysis by hypertension status at baseline.

Author Response

Thank you in advance for your kind and useful comments. Here we send our response and revised manuscript according to your suggestions. We really appreciate that your comments substantially improved the scientific validity of our study. We will look forward to hearing your positive reply soon. Thanks.

Reviewer 1

Comments and Suggestions for Authors

I have no further comments. Based on the limited sample size, the authors did not have enough power to stratify the association by hypertension status.

If this paper was considered for publication, as I mentioned in my last comments, two points must be acknowledged clearly in the discussion part: (1) Because the baseline SBP level in the PPH group was much higher than that in the Non-PPH group, the association observed between PPH and incident CVD may only reflect the high baseline SBP on risk of CVD. Although the authors adjusted hypertension in the multivariate model, based on the small sample size, I suppose some effect of baseline blood pressure level is still there. (2) Due to the limited sample size, the authors have no ability to stratify the analysis by hypertension status at baseline.

Response: Thank you for your kind and useful comments that helped us to improve the manuscript. I believe your comments substantially improved the scientific quality of the study. I understand an agree with your concern regarding reader comprehension. As you pointed out, we believe that a study with a larger sample size will be required to exclude the possible effect of hypertension on relationship between PPH and CVD. We believe this point should be clearly acknowledged to the readers. Therefore, I have revised the limitations in the Discussion section and have clarified the point on small sample size (Discussion, Lines 259-265):

Fifth, the positive association between PPH and CVD may only reflect the high baseline SBP on the risk of CVD because the baseline SBP level in the PPH group was much higher than that in the non-PPH group. Although hypertension was adjusted in the multivariate model, based on the small sample size, the effect of high baseline SBP may persist. As it is impossible to stratify the analysis by hypertension status at baseline owing to the limited sample size, further research in the form of a well-designed, large-sample study in a normotensive elderly population is necessary to definitively establish the effect of PPH on the risk of CVD.

Reviewer 2 Report

None

Author Response

Reviewer 2

Comments and Suggestions for Authors

None

Response:

Dear reviewer,

Thank you for accepting this manuscript. I believe you’re your previous comments helped us to improve the clarity of the manuscript. Your inputs are appreciated.

This manuscript is a resubmission of an earlier submission. The following is a list of the peer review reports and author responses from that submission.

Round 1

Reviewer 1 Report

The authors investigated the association between postprandial hypotension (PPH) and risk of incident CVD in elderly people, and concluded that the existence of PPH can predict the development of new CVD. However, before some key questions were dealt with, the conclusion is questionable. My comments are as follows.

Major comments

The distribution of baseline characteristics in participants with PPH and without PPH is not comparable or balanced in some key variables, such as prevalence of hypertension, diabetes and educational level. The “no significant differences in baseline characteristics between participants with and without PPH” is just caused by limited sample size that the authors had no power to detect differences. What about the changes of baseline characteristics during the three-year follow-up, i.e., the time-varying covariates. It is possible that participants with PPH had higher blood pressure level at baseline (the authors did not provide the mean and SD of baseline blood pressure), and most of them may have developed to hypertension during follow-up. The authors did not consider the time-varying covariates, such as blood pressure, glucose level, BMI level, etc. The authors did not mention anti-hypertension treatment. If participants took antihypertensive drugs before meal, the decreased blood pressure after meal cannot be attributed to PPH, and this may cause misclassification of participants. Did the authors consider physical activity in their model? Did the two groups have same level of physical activity levels? The authors said “CVD-related information was obtained from them or their families during their visits to the centers or through telephone calls”. This means the CVD endpoints were collected by self-report, and were not validated by hospital records.

Minor Comments

Please reported the baseline blood pressure level in two groups in mean (SD) format. “During the follow-up, seven participants died of coronary heart disease (n=2), cerebrovascular 108 disease (n=2), pneumonia (n=2), and sepsis (n=1), while 30 (31.9%) developed new CVD.” This sentence is not clear, please rephrase it. The authors mentioned “investigated the risk of developing CVD in healthy elderly people with PPH”. Actually, they are not “healthy”, since they may have hypertension and diabetes at baseline.

Reviewer 2 Report

The objective of this study is to examine whether CVD development depends on the PPH. A total of 96 older adults were enrolled in this study. The results have shown statistical relationship between PPH and CVD development. The authors conclude that PPH can predict the development of new CVD.

The topic is a clinically significant and interesting. However, since sample size was very limited, the statistical analyses in this study were insufficient. This is very important points. The increase in number of the subjects should be essential in this study. The pathogenesis still remains unclear. Why did PPH influence to the CVD? I can’t understand the biological relationship. In addition, confounding factors were not considered. Authors should perform a multivariate analysis among the significant factors with 2 variable analysis or adjust the confounding factors statistically so that they can extract factors affecting CVD development independently.

Minor points:

Did the author confirm the inter-examiner error? Authors need to indicate if the examiners were standardized and how many examiners performed the clinical exams. Why did author used Student’s t-test to compare continuous variables? Were the data normally distributed?